# The first transmembrane region of complement component-9 acts as a brake on its self-assembly

Bradley A. Spicer[1,2], Ruby H.P. Law[1,2], Tom T. Caradoc-Davies[1,3], Sue M. Ekkel[1,2], Charles Bayly-Jones[1,2], Siew-Siew Pang[1,2], Paul J. Conroy [1,2], Georg Ramm [1,2], Mazdak Radjainia[4], Hariprasad Venugopal[1,2], James C. Whisstock[1,2,5,6] & Michelle A. Dunstone[1,2]

Complement component 9 (C9) functions as the pore-forming component of the Membrane Attack Complex (MAC). During MAC assembly, multiple copies of C9 are sequentially recruited to membrane associated C5b8 to form a pore. Here we determined the 2.2 Å crystal structure of monomeric murine C9 and the 3.9 Å resolution cryo EM structure of C9 in a polymeric assembly. Comparison with other MAC proteins reveals that the first trans-membrane region (TMH1) in monomeric C9 is uniquely positioned and functions to inhibit its self-assembly in the absence of C5b8. We further show that following C9 recruitment to C5b8, a conformational change in TMH1 permits unidirectional and sequential binding of additional C9 monomers to the growing MAC. This mechanism of pore formation con-trasts with related proteins, such as perforin and the cholesterol dependent cytolysins, where it is believed that pre-pore assembly occurs prior to the simultaneous release of the transmembrane regions.

[1] ARC Centre of Excellence in Advanced Molecular Imaging, Department of Biochemistry and Molecular Biology, 23 Innovation Walk, Monash University, Victoria, 3800 Australia. [2] Biomedicine Discovery Institute, Department of Biochemistry and Molecular Biology, 23 Innovation Walk, Monash University, Victoria, 3800 Australia. [3] Australian Synchrotron, 800 Blackburn Road, Clayton, Victoria, 3168 Australia. [4] Achtseweg Noord 5, Building, 5651 GG Eindhoven, The Netherlands. [5] EMBL Australia, Monash University, Melbourne, Australia. [6] South East University-Monash Joint Institute, Institute of Life Sciences, Southeast University, 210096 Nanjing, China. These authors contributed equally: Bradley A. Spicer, Ruby H. P. Law. Correspondence and requests for materials should be addressed to J.C.W. (email: James.Whisstock@monash.edu) or to M.A.D. (email: Michelle.Dunstone@monash.edu)

The MAC represents the terminal portion of the complement system, and functions to form large pores in the membrane of target bacteria, enveloped viruses and parasites[1,2]. Currently, it is suggested that the MAC assembles on the target membrane via sequential addition of five different components (C5, C6, C7, C8 [comprising C8α C8β, and C8γ], and C9). The final stage of MAC formation involves addition of multiple copies of the pore-forming component, C9, to the C5b8 complex. Together these proteins form an asymmetric pore (Supplementary Fig. 1)[3,4].

C6, C7, C8α, C8β, and C9 all contain a membrane attack complex/perforin/cholesterol dependent cytolysin (MACPF/CDC) domain (Supplementary Fig. 2). This domain is generally associated with a pore forming function in a wide variety of different toxins and immune defence proteins[5]. Previous work reveals that the mechanism of MACPF/CDC pore formation involves three steps[6–8]. First, soluble monomers are recruited to the membrane. Next, between 10 and 50 membrane associated molecules then laterally migrate and self-assemble into a circular or arc pre-pore form[8]. Finally, a conformational change in two regions (named TMH1 and TMH2 because of the β-hairpin conformation each region finally adopts in the membrane) results in formation of an unusually large, membrane spanning β-barrel pore (Supplementary Fig. 1)[9,10]. Each subunit contributes two membrane spanning β-hairpins. Further, based on AFM studies of CDCs and structural studies on the fungal MACPF toxin pleurotolysin, it is postulated that membrane insertion of all membrane spanning regions occur in a simultaneous fashion[8,11].

Most MACPF/CDC proteins, such as CDCs[8] and perforin[12], involve a single protein that can self-associate, usually only in the context of having bound first to a lipid membrane. In contrast, the MAC is unusual in that it is initiated by a non-MACPF domain protein, C5b, which then allows the sequential binding of single units of the MACPF-domain containing proteins C6, C7 and C8 complex (C8αβγ). This assembly (C5b8) then allows the binding of multiple units of C9 that form the final ring-shaped pore (Supplementary Fig. 1a). C9 can also form a homogenous ring in vitro, called polyC9, that closely resembles the assembly of C9 in the MAC[13].

Previous studies reveal that the MAC assembles via sequential recruitment of each component from the soluble phase onto the growing, membrane associated complex in a unidirectional manner[14]. Each component of the MAC thus contains a binding surface and an elongation surface (Supplementary Fig. 1b). Once an individual component is associated with the nascent MAC, its elongation surface is activated (presumably via a conformational change) such that it can now interact with the binding surface of the next soluble component to join the complex (Supplementary Fig 1). The final component of the MAC, C9, is the only component of the assembly that can self-associate—an event that completes the structure of the pore. The complete MAC contains ~18 C9-monomers in the full assembly.

Currently, given the ability of C9 to self-associate, it is unknown how aberrant oligomerisation is prevented in the solution phase prior to binding the C5b8 complex. To address this question, we determined the 2.2 Å X-ray crystal structure of monomeric murine C9 together with the 3.9 Å resolution cryo EM structure of poly-C9. These data show that TMH1 functions in monomeric C9 to sterically inhibit its self-assembly in the absence of C5b8. Binding of C9 to C5b8 results in a conformational change in TMH1 and entry of this region into the membrane. This transition, which is likely accompanied by release of TMH2 and movement in a conserved helix-turn-helix (HTH) structure, permits sequential and unidirectional recruitment of the next C9 monomer into the growing MAC.

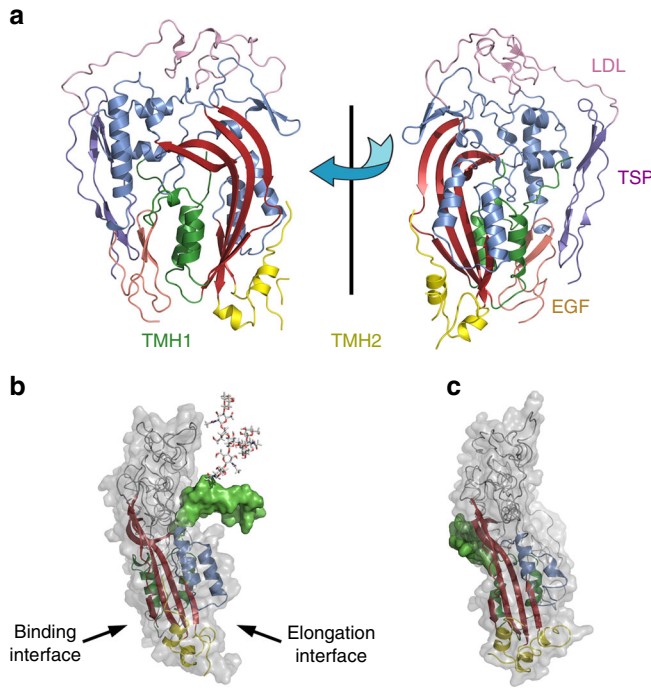

**Fig. 1** The X-ray crystal structure of complement component 9. **a** The X-ray structure of C9 shown in cartoon in two orientations, rotated 180° apart. The bent β-sheet of the MACPF domain is shown in red with α-helices in blue, TMH1 (green) and TMH2 (yellow). The ancillary domains: TSP1 (purple), LDLRA (pink) and EGF domain (orange). Domain colours also match the colours used to show the domain features in Supplementary Fig. 2. **b** Cartoon model of C9 with the modelled TMH1 loop (green surface) and N-glycan (PDB ID 1HD4) located on the elongation face of the protein. The key features of the MACPF domain are shown as cartoon and coloured as follows: central β-sheet (red), TMH 1 (green), TMH2 (yellow), HTH (blue). **c** The C8β structure in the same orientation as C9 showing the TMH1 domain on the docking interface (PDB ID 3OJY)

## Results

**The X-ray crystal structure of monomeric C9.** The structure of C9 reveals that the four domains (Thrombospondin type 1 [TSP1], Low density Lipoprotein Receptor Type A [LDLRA], MACPF/CDC and Epidermal growth factor [EGF], Supplementary Fig. 2) are arranged into a globular bundle. Structural comparisons with other MAC proteins (e.g., C8β) reveal that the overall arrangement of domains is similar (Supplementary Fig. 3) except for a striking difference in the position of TMH1 with respect to the core body of the molecule (Fig. 1b, c)[14,15].

In the structure of C6, C8α, and C8β, TMH1 is arranged such that it does not obviously obstruct binding to the next subunit. Indeed, the structure of the complex between C8α and C8β reveals that the TMH1 of the monomeric C8α is buried within the interface[15]. The interface of each of these proteins is relatively flat and currently the precise nature of the conformational changes that take place in order for a new molecule to join the assembly remains to be completely understood. In these regards, we and others, have suggested that a structurally conserved Helix-Turn-Helix motif that sits on top of TMH2 represents the major component that must shift during pre-pore assembly[11,13,16,17].

The monomeric C9 crystal structure reveals that a large proportion of TMH1 (a portion of which is flexible and cannot be resolved in electron density) is located in the centre of the elongation surface where it would be anticipated to block binding

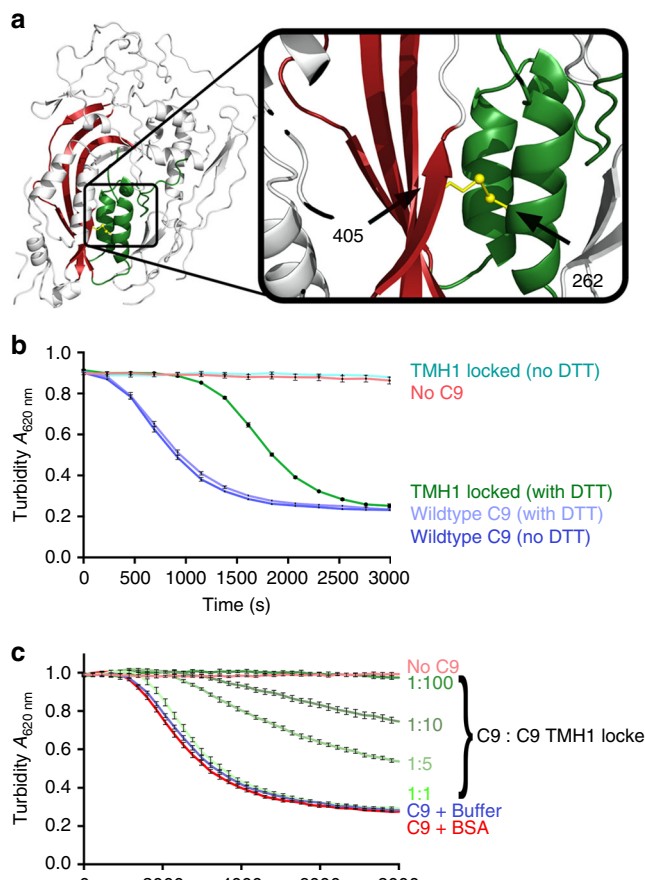

**Fig. 2** The C9 TMH1 movement is necessary for pore assembly. **a** Cartoon model of a C9 monomer (left) disulphide locked mutant (also called C9$_{mutant}$ with F262C/V405C mutations) (shown as yellow sticks), that links the TMH1 region to β-strand 4 of the MACPF domain (right). **b** Haemolytic activity of disulphide locked C9 against erythrocytes/antibody/complement 1–8 (EAC1-8). The TMH1 locked (no DTT) alone is inactive; however, activity can be rescued with 1 mM DTT (TMH1 locked (with DTT)). Also shown are control experiments: no C9, and wildtype C9 (with and without DTT). **c** Competition assay of disulphide locked mutant with wildtype C9 showing that the disulphide trapped variant competes for the elongation face with wild-type C9. A range of ratios of wildtype C9 and C9 TMH1 locked mutant used in the assays are as shown and it reveals that the disulphide locked C9 competes for the nascent MAC and stalls assembly in a dose-dependent manner. Also shown are no C9, C9 in buffer and C9 plus BSA controls. The results (**b** and **c**) are presented as the averaged turbidity measurements from three independently prepared samples (n = 3) with error reported as the standard error of the mean (SEM). See also Supplementary Fig. 7 for more detail

of another C9 monomer (Fig. 1b). Interestingly, the flexible region of TMH1 is the least conserved region across all vertebrate species and thus may represent a site under significant evolutionary pressure, for example, as a site of MAC inhibition by bacteria[18]. In addition, it is notable that TMH1 contains an N-glycan (found in most species) on residue N243 (human equivalent N256), a modification that would add additional bulk to this region. This finding suggests that TMH1 may function to block self-assembly, and that a key event of the interaction between C9 subunits would be a conformational

change of TMH1 such that it moves to reveal the C9 elongation surface.

**Mobility in TMH1 is essential for C9 self-association**. To investigate the hypothesis that TMH1 blocks self-assembly, we designed a disulphide trap mutant (F262C/V405C, [C9$_{mutant}$]) that linked TMH1 to β-strand 4 of the MACPF/CDC domain (Fig. 2a). Time resolved haemolytic assays revealed that the disulphide-trapped C9 variant is completely inactive with respect to lytic function and that addition of reducing agent resulted in restoration of lytic function (Fig. 2b). Crucially competition assays further reveal that the C9$_{mutant}$ competes with wild type C9 and thus is competent to bind the C5b8 or C5b89$_n$ complex to form C5b89$_{mutant}$ or C5b89$_{n+mutant}$ respectively (Fig. 2c). Together these data suggest that the sequential addition of C9 molecules to C5b89$_1$ relies on a rearrangement in TMH1.

**The cryo-EM structure of poly C9**. To further investigate the structural transitions associated with C9 self-assembly, we determined the 3.9 Å resolution cryo EM structure of polyC9 (Fig. 3a), the highest resolution structure to date of any MACPF or CDC protein in the pore form. PolyC9 mimics the form seen in the complete MAC[13] and is formed in vitro following prolonged incubation of concentrated C9 at 37 °C.

The resolution of the polyC9 map ranges from 3.2 to 4.4 Å, with the best resolution present at the top of the β-barrel around the HTH region (Supplementary Figs. 4, 5). As previously reported from analysis of our lower resolution (8 Å) structure[13], the final structure of human polyC9 reported in this study contains 22 monomers. The improved resolution of the maps permitted construction and refinement of a full atomistic model. In this model, we were able to unambiguously assign 460/528 residues of main-chain atoms. Clear electron density was observed for side chains located at the oligomer interface and around the HTH region (described below). Further, and in regards to the remarkable 88 stranded β-barrel itself, our data were of sufficient quality to reveal individual strands within the assembly, providing experimental evidence that the β-barrel adopts the $S = n/2$ architecture as predicted by bioinformatics analysis[19].

Analysis of the polyC9 model revealed charge complementarity between the elongation face and binding face of each subunit (Supplementary Fig. 6). A total of 91 contacts are made at the interface between subunits (Supplementary Table 1). Six of these interactions involve the TSP1 domain, which plays an important role in pore assembly and is intercalated around the outer edge of the ring[13,20].

**The molecular transitions that control MAC assembly**. A comparison to the polyC9 structure with the monomeric C9 form revealed that TMH1 must move to expose the elongation face for an additional C9 monomer to bind to the growing assembly (Fig. 3b). In addition, these structural comparisons reveal that the HTH region must also be substantially repositioned to permit binding of the next C9 subunit.

In the monomeric structure of C9 the HTH packs against the underlying β-sheet as well as part of TMH2 (Fig. 3c). In polyC9, however, both TMH1 and TMH2 are released and, as a consequence, the HTH region has moved such that it partially occupies the position vacated by TMH2 (Fig. 3c). Analysis of side chain interactions revealed that, in both the monomer and pore structures the HTH is loosely packed against the surrounding structures, and makes mainly hydrophobic bonds (e.g., Fig. 3d, e). These data are consistent with this region being able to readily move in response to conformational change in TMH2. The HTH

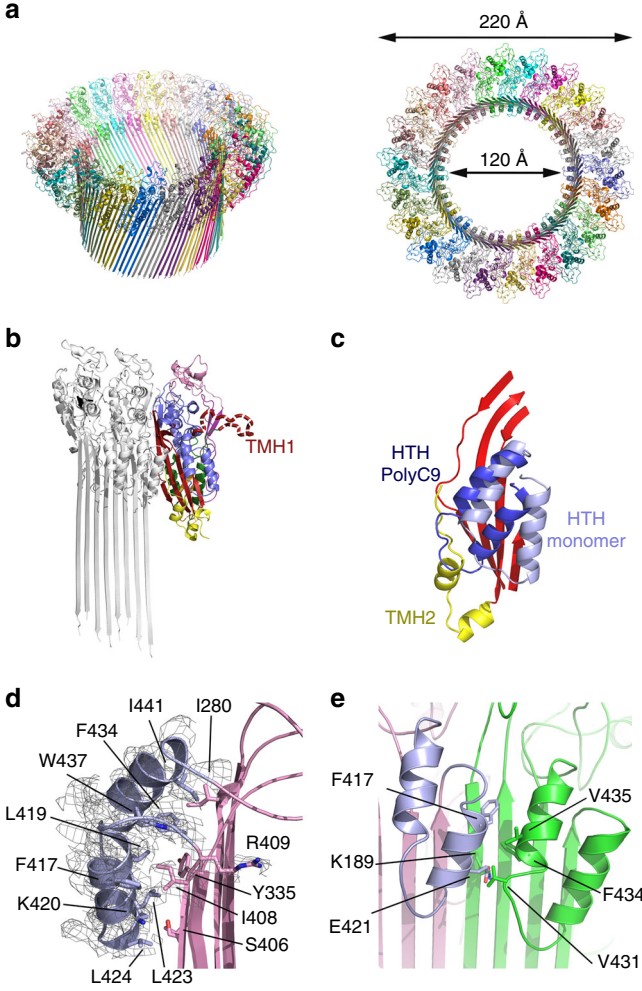

**Fig. 3** C9 structure in the monomeric and assembled forms. **a** The cryo EM structure of polyC9 with 22 subunits (different colours) in a circular assembly, two orientations shown, oblique-view and top-down. The resolved strands in the β-barrel conform to the $S = n/2$ architecture[19]. The model excludes the membrane spanning region (Supplementary Fig. 5a). **b** Cartoon representation of C9 monomer (colour) docked to a previously unfurled assembled C9 dimer (grey, left). The position of TMH1 as well as the HTH blocks the elongation face. **c** Relative positions of the HTH of polyC9 (dark blue) and monomeric C9 (light blue). The central β-sheet of the MACPF domain is also shown (red). In polyC9, the HTH partially occupies the region vacated by the TMH2 α-helices (yellow). **d** Zoom in view of the HTH domain in the polyC9 EM map, key residues found in the interface between HTH and β-barrel are shown in sticks and labelled. **e** Cartoon representation of two HTH regions, plus key residues in the interface, from neighbouring C9 molecules in the polyC9 structure

further makes new inter-subunit interactions in polyC9 (Fig. 3e) such that it forms a continuous band of α-helices that line the top of the β-barrel lumen (Fig. 3a). Outside of these regions, and in the context of the elongation face of C9, only modest shifts of individual rigid bodies and domains are required to permit MAC polymerisation (Supplementary Table 2), for example, the TSP1 translates with respect to the EGF domain by ~2 Å.

## Discussion

Previous data suggest that both TMH1 and TMH2 of C8 fully enter the membrane prior to recruitment of C9[21]. This finding is consistent with the observation that incomplete arc-like structures can form and penetrate the membrane[4,8]. In such a structure (where C8 is fully inserted) the edge strand at the elongation face is the TMH2 of C8α (Supplementary Fig. 1). Our biochemical data reveal that a C9 variant in which TMH1 is disulphide trapped is able to join C5b8, however, further elongation is not possible without the release of TMH1. We hypothesise that upon binding to C5b8 the most likely next step is for the TMH1 of C9 to add to the nascent barrel structure by forming a canonical β-hairpin with the membrane inserted TMH2 of C8α (Fig. 4). Alternatively, it is possible that TMH1 moves sufficiently to permit C9 binding, but without inserting into the membrane. However, we have no evidence for such an intermediate pre-pore like state.

We further suggest that prior to the next C9 subunit joining the assembly, it is highly likely that the TMH2 of C9 is also released to enter the membrane, and that this permits the HTH region to slide across the underlying β-sheet. The removal of TMH1 together with the shift in the HTH region will expose the elongation face of C5b89₁ and permit recruitment of the next C9 subunit into the growing MAC. Taken together these data explain how the MAC has evolved a mechanism of coupling sequential insertion with elongation. The mechanism of C9 pore formation also directly contrasts that of related molecules such as perforin and the CDCs, where it is suggested that the assembly of pre-pores (or pre-pore-like arcs) takes place prior to the simultaneous release of the membrane spanning regions[6–8,12,22].

## Methods

**Recombinant C9 purification.** Human C9 and mouse C9 protein were purified using similar methods (with minor variations). The human C9 gene (P02748) was cloned into pSectag2a (Thermo Fisher Scientific) for expression in mammalian Expi293 cells where the native secretion sequence was replaced with the Igκ leader sequence. Human C9 mutants (F262C, V405C and F262C/V405C) were cloned using QuikChange. The mouse C9 (P06683) sequence was synthesised and cloned into pcDNA3.1 vector (GeneScript) also containing an Igκ leader sequence. Recombinant protein was produced by transient expression in Expi293F cells (Thermo Fisher Scientific) for four days according to the manufacturer's instructions. The oligonucleotide primers used for cloning can be found in Supplementary Table 3.

The purification methods were essentially the same as previous one[13,23] with some exceptions. Following centrifugation, the Expi293 media containing C9 was diluted with an equal volume of 10 mM sodium phosphate pH 7.4, 20 mM NaCl containing cOmplete protease inhibitor tablet (Roche). Then, it was loaded onto an equilibrated, HiTrap DEAE sepharose column (1 mL resin per 100 mL media). Chromatography steps were performed on an ÄKTA FPLC. The protein was eluted from the DEAE column using a linear gradient over 20 column volumes (from 10 mM sodium phosphate, 45 mM NaCl, pH 7.4 to 10 mM sodium phosphate, 500 mM NaCl, pH 7.4). Pooled fractions containing C9 were further purified using hydroxyapatite specifically using a pre-packed Bio-Rad type I CHT column equilibrated in 10 mM sodium phosphate pH 7.0, 100 mM NaCl. The CHT elution was performed over a six column volumes phosphate gradient at pH 8.1 (from 45 mM to 350 mM). Pooled fractions were concentrated using a 30 kDa MWCO concentrator (Amicon) and further purified using a size exclusion column; prepacked Superdex S200 16 mm × 60 mm or 26 mm × 60 mm (GE Healthcare life sciences). Size exclusion chromatography for human C9 was performed in 10 mM HEPES pH 7.2, 200 mM NaCl whereas mouse C9 was purified in 10 mM HEPES pH 7.2, 100 mM NaCl.

**Murine C9 crystallisation.** The murine C9 gene was synthesised (GenScript) with three mutations (N28E; N243D and N397D) in order to produce a non-N-linked glycosylated protein for crystallisation trials. The modified C9 was consistently observed to have similar activity to human C9 in a haemolytic assay using sheep erythrocyte/antibody/complement 1–8 (EAC 1–8, human C9-depleted serum; ComplementTech). Recombinant murine C9 was purified as described above, and the C9 protein sample was concentrated to ~9 mg mL⁻¹ (~150 μM) for crystallisation trials. Optimised crystals were obtained using the hanging drop vapour diffusion method with a reservoir liquor containing 18% (w/v) PEG 3350, 0.2 M disodium malonate pH 7.5 and 10 mM ZnCl₂ using the micro-seeding method. Crystals were flash cooled in liquid N₂ with 25% (v/v) glycerol as a cryoprotectant. Data collection was performed at the Australian Synchrotron MX2 Beamline. Experimental phasing of C9 crystals was performed by soaking crystals in tantalum bromide (Jena Biosciences) and uranyl formate (Polysciences, Inc) for two days prior to harvesting.

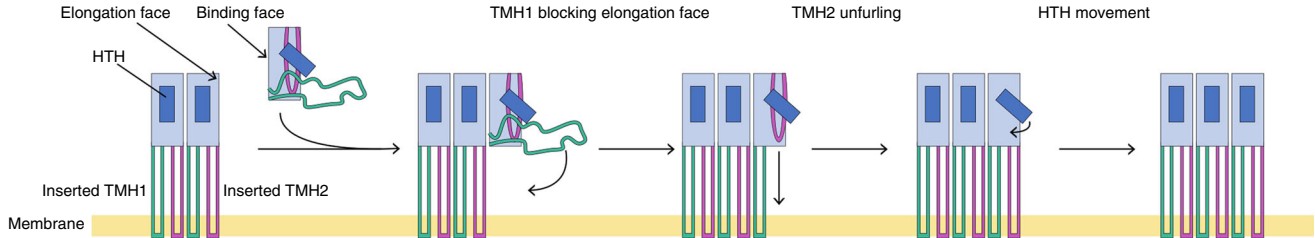

**Fig. 4** Schematic diagram of the unidirectional C9 assembly. We hypothesise that during the assembly, binding to the elongation face of a C9 subunit leads to the release of TMH1 which inserts to form a canonical β-hairpin. Following this, the release of TMH2 and a conformational change in the HTH region uncovers the elongation face of the newly assembled C9, allowing the next C9 subunit to join the assembly

**X-ray data collection and model building**. The data were merged and processed using XDS[24,25], POINTLESS[26,27] and AIMLESS[28]. Five percent of the datasets were flagged as a validation set for calculation of the $R_{free}$ with neither a sigma, nor a low-resolution cut-off applied to the data. Experimental phases (Supplementary Table 4) were obtained by the MIRAS (multiple isomorphous replacement plus anomalous differences) method. Molecular replacement was attempted using the MACPF domain of C6 (PDB ID 3T5O) and with both MACPF domains of C8 (PDB ID 3OJY). None of the MR experiments were successful. The Ta and U heavy atoms were not ordered and the structure was phased using the anomalous signal of the Zn and Ca ions bound to the protein. Two datasets collected at 10,300 eV (which is above both the K-edge of Zn and L-III edge of Ta), were used as derivative 1 and derivative 2 datasets (Supplementary Table 4). Experimental phasing strategies and dataset combinations were evaluated using HKL2MAP[29] and final phasing was carried out using the CRANK2 pipeline[30]; heavy atom positions were located using SHELXC/SHELXD[31], substructure refinement was done using BP3[32]. The initial FOM (figure of merit) from phasing was 0.26 and after density modification with PARROT[33] this increased to 0.57. Automated model building was carried out using BUCCANEER[34] with the initial model consisting of 944 residues with $R/R_{free}$ of 34.1/40.0%. Two molecules were found per asymmetric unit. Model building was performed using COOT[35] while refinement was performed using PHENIX[36], REFMAC[37], and autoBUSTER[38]. Water molecules were added to the model when the $R_{free}$ reached 30%. Crystallographic and structural analysis was performed using CCP4 suite[39] unless otherwise specified. All Zn atoms were modelled into the omit-map generated using ANODE[40] from a dataset collected at 9674.0 eV (1.28162 Å), above the K-edge of Zn (9659.0 eV), and confirmed by the absence of anomalous signal at the Zn sites in a dataset collected below the Zn K-edge at 9643.9 eV (1.28562 Å) (Supplementary Table 5). Figures 1a, b, and 2a; Supplementary Fig. 3; Supplementary Fig. 4 were generated in part using PYMOL and Chimera[41]. The final model contains two chains: chain A is less flexible with residues 18–226, 248–365, 395–526 modelled into the electron density; chain B residues 18–73, 78–113, 116–205, 214–225, 249–364, and 395–526 modelled. In the final model, the number of residues in the Ramachandran favoured region is 873 residues (out of a total of 874 residues). Structural validation was performed using MolProbity[42]. The MolProbity score is 0.87 which is in the 100th percentile of structures reported at this resolution.

**PolyC9 preparation and data collection**. Mammalian cell expressed human C9 (with the two native N-glycans) was buffer exchanged by dialysis into 10 mM HEPES pH 7.5, 50 mM NaCl overnight at 4 °C at a concentration between 100 and 250 μg mL$^{-1}$. Following dialysis, the human C9 was concentrated between 1.1–1.5 mg mL$^{-1}$ with a 30 kDa MWCO (Amicon) protein spin filter and 1:9 (v/v) of amphipol A8-35 (Anatrace) was added to a final concentration of 0.015–0.02 mg mL$^{-1}$. Polymerisation reactions were initiated by incubating at 37 °C overnight and stored at 4 °C.

The polyC9 reaction producing the best grid was from an initial C9 concentration of 1.3 mg mL$^{-1}$ containing 0.02 mg mL$^{-1}$ A8-35. Plunge-freezing was performed using a Vitrobot Mark IV (FEI/Thermo Fisher Scientific). PolyC9 (2.5 μL) was added to a freshly glow discharged Quantifoil copper grid (R1.2/1.3, 200 mesh). Data was collected on a Titan Krios (FEI/Thermo Fisher Scientific)) operated at 300 kV at a magnification of 130 K in microprobe EFTEM mode, resulting in a magnified pixel size of 1.06 Å pixel$^{-1}$. The movies were collected using a Gatan K2 Summit with a quantum energy filter in super resolution mode (for an effective pixel size of 0.53 Å pixel$^{-1}$). Each movie consists of 20 sub frames and the exposure time was 8 s which amounted to a total dose of 46.4 e$^-$ Å$^{-2}$ at a dose rate of 6 e$^-$ Å$^{-2}$ s$^{-1}$.

**Cryo-EM data processing**. Unless stated otherwise all processing was performed with RELION (v2.1b.1)[43]. Movies were down sampled in Fourier space by a factor of 2 and summed after correction of beam-induced motion by Motion-Cor2[44]. CTF estimation was performed by CTFFIND4.1[45] and micrographs with

ice contamination were discarded by visual inspection of the power spectra. Initially ~1000 particles were manually picked and subjected to reference-free 2D classification to serve as templates for auto picking. A total of ~220,000 particles were extracted from summed micrographs and subjected to multiple rounds of 2D classification. A representative subset of class averages was selected for initial model generation in EMAN2.2[46] using the common line method. The initial model was low pass filtered to 20 Å and particles were subjected to 3D classification, giving rise to two classes of C22 and C21 symmetry. Initial refinement with C22 symmetry led to a 4.2 Å map. These initial refinements were used to create a solvent mask, which was low pass filtered to 15 Å for subsequent refinements. This final subset of 58,000 particles was selected for masked movie refinement and particle polishing with C22 symmetry, where the MTF of the detector was used to determine a $b$-factor of $-180$ Å$^2$. High resolution features were enhanced by sharpening with this $b$-factor for the purposes of map visualisation. The global resolution was estimated by the Gold Standard 0.143 criterion when comparing the Fourier shell correlation between two independent half maps[47]. The local variation of resolution was further analysed using blocres using a search box size of 20 voxels and FSC criterion of 0.5[48].

**PolyC9 model building and model validation**. Model building of the polyC9 was performed in COOT (0.8.8)[49]. The crystal structure of murine C9 was manually positioned into the best density of the cryo-EM map and rigid body fitting of individual domains was performed. The TMH1 and TMH2 regions, which significantly alter conformation, were removed for manual building. Non-conserved amino acids were mutated from murine to human residues and their side chains manually positioned to maximise fit in the map. Following initial model building, C22 symmetry was applied to the single subunit of polyC9 using Chimera (UCSF, USA)[50] and further real space refinement performed in COOT to minimise clashes between subunits and improve the overall geometry.

The final three-dimensional model of polyC9 was refined into the cryo-EM map using the phenix.real_space_refine programme within PHENIX suite to optimise and correct for poor geometry (Supplementary Table 6)[36]. During the refinement, standard restrains for covalent geometry, Ramachandran plot and internal molecular (NCS) symmetry were imposed. In addition, secondary-structure restrains were defined for the β-barrel region of the pore (β-strands 186–216, 251–281, 333–363, 379–409) because the map quality towards the end of the pore is of lower resolution. Protein Interactions Calculator[51] was used to calculate intermolecular contacts between adjacent molecules of polyC9 (Supplementary Table 1).

**Haemolytic assay**. Turbidity measurements were performed using sheep EAC1-8 prepared in DHB++ pH 7.4 (Dextrose HEPES Buffer; containing 2.5% (w/v) D-glucose, 5 mM HEPES, 71 mM NaCl, 0.15 mM CaCl$_2$, 0.5 mM MgCl$_2$). EAC1–8 was produced by sensitising $6.5 \times 10^8$ cells mL$^{-1}$ sRBC with equal volume of anti-sheep antibody (0.75 mg mL$^{-1}$) (Rockland immunochemicals cat no. C220–0002) at 30 °C. Sensitised cells were washed 2 min at 3220×g by centrifugation, and then C9-depleted serum (Complement Tech) added in batch with 1 μL per $3.75 \times 10^6$ cells and incubated at 37 °C for 30 min. The absorbance at 620 nm was continuously measured while incubating at 37 °C with intermittent orbital mixing. For unlocking experiments three independently prepared dilutions of C9 were combined with EAC1–8 ($3.75 \times 10^6$ cells) and a final concentration of 1 mM DTT in a 96-well plate. Competition assays were prepared by combining different ratios of the TMH1 locked C9 (F262C/V405C, [C9$_{mutant}$]) with a constant amount of wild-type C9 (final concentration 270 ng mL$^{-1}$) prior to addition of EAC1-8 or BSA which was used as a non-specific binding control. Data are reported as the raw turbidity curves with error reported as the standard error of the mean.

**Data availability**. Data supporting the findings of this manuscript are available from the corresponding authors upon reasonable request. The datasets generated

during the current study are available in the RCSB repository (PDB ID 6CXO) and (PDB ID 6DLW) and the EMDB repository (EMDB ID 7773).

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

## Acknowledgements

J.C.W. is a National Health and Medical Research Council of Australia (NHMRC) Senior Principal Research Fellowship and acknowledges the previous support of an Australian Research Council (ARC) Federation Fellowship. M.A.D. is an ARC Future Fellow. B.A.S. is supported by an MBIO PhD Scholarship. C.B.J. is supported by an Australian Government RTP Scholarship. We thank the ARC and NHMRC for funding support. We thank the Monash platforms (crystallisation, protein production, the Monash Clive and Vera Rama-ciotti Electron Microscopy Centre, Proteomics, and eResearch [the MASSIVE super-computer] for technical support. This research was undertaken in part using the MX2 beamline at the Australian Synchrotron, part of ANSTO, and made use of the ACRF detector.

## Author contributions

M.A.D. and J.C.W. conceived the study, co-led the work and co-wrote the paper. B.A.S., R.H.P.L., T.T.C.D. collected data, determined the structures and co-wrote the paper. B.A.S.,

S.M.E., C.B.J., S.S.P. and P.J.C. produced and analysed protein. M.R., G.R. and H.V. setup collection of EM experiments.

## Additional information

**Competing interests:** The authors declare no competing interests.

