## [Peer Review File · Nature Communications]

Reviewers' comments:

Reviewer #1 (Remarks to the Author):

This manuscript by Spicer, Law et al. describes structural studies of monomeric C9 by crystallography and pore-forming polyC9 by cryo-EM. The structural work is done beautifully and the authors are congratulated with their important achievements.

However, I find it hard to follow their reasoning in terms of a mechanistic model in the sequential process of adding a monomer to the growing ring of polyC9. One difficulty lies in Fig. 1 b and c. The text mentions a "striking difference". This is simply not clear to me from Fig. 1. Figure 3c does to some degree a better job of showing how the structure alters upon polymerisation.

Next, an important aspect of the mechanistic model is the disulphide trapped variant of C9, which locks TMH1. Though, by itself the resulting data are interesting, the data could be interpreted in multiple ways. Clearly, a variant that is unable to form the two beta-hairpins presents in possibly multiple ways a corrupted ending making it harder to bind a next copy. To show that a structural element in monomeric C9 prevents polymerisation, would require mutants in which this element is removed or altered such that polymerisation proceeds more readily.

Nevertheless, the structural data themselves present hallmarks to the field. I would suggest to improve, in particular, Fig 1b/c and rephrase the manuscript w.r.t. its mechanistic interpretation.

Minor comments:

The abstract reports 2.3 angstrom resolution for the X-ray structures, whereas in the text this is 2.2 angstroms.

The gamma character is not correctly printed in the supplement file.

Reviewer #2 (Remarks to the Author):

This manuscript describes the first high-resolution x-ray crystal structure of complement C9 and additionally a high-resolution cryo-EM structure of polyC9. Lack of C9 structural information has been a major obstacle to understanding details of how the MAC assembles and interacts with membranes. The authors work is highly significant in that it extends previous hypotheses about MAC assembly based on C6 and C8 structures and it provides new insight into how C9 can exist independently in monomer form yet self-polymerize when associated with the MAC. The authors' conclusions regarding the role of TMH1 in preventing C9 self-polymerization are plausible based on differences from structures of C6, C8a and C8b. Importantly, these conclusions are supported by experiments using mutants in which TMH1 is locked to a core b-strand. Determining the C9 structure alone is a major advance in the complement field but equally important are the mechanistic implications with respect to understanding polyC9 and MAC formation. These results will have a major impact on the area of complement.

The authors should identify the species of C9 as "mouse" in the title and/or abstract.

Reviewer #3 (Remarks to the Author):

The manuscript by Spicer et al. describes the monomeric X-ray structure of the immune effector Complement component 9 (C9) at 2.3Å and the cryo-EM structure of the polymeric C9 assembly (polyC9) of the membrane attack complex (MAC) at 3.9Å resolution. By comparison of the monomeric X-ray structure to the cryo-EM structure of the polymeric C9, the authors reveal molecular movements of important regions (HTH) in the molecule that appear to be required for assembly of the pore complex and subsequent pore formation. By mutating opposing residues in

the central beta-sheet region and "TMH1" to cysteines and crosslinking them as disulfide bridges in the absence of DTT they verified that the movements they propose are required for functionality of C9 to form the pore complex.

The authors performed their experiments properly and present their results very briefly. Nevertheless, the conclusions were supported by the data. The manuscript fulfills all the requirements and therefore suitable for publication in Nature communications after the following points are addressed, clarified or discussed.

As the text structure suggests, the manuscript seems to be written very condensed in the introduction and results (for Nature?). The discussion is somehow squeezed into the text, which to my knowledge is not necessary for Nature communications. Therefore the manuscript would benefit from an expansion and broader description and discussion of the results.

Abstract:

Last sentence of the abstract: How the authors distinguish the MAC pore formation by sequential addition of C9 monomers from that of the CDC pore formation by the "growing-pore mechanism" where the CDC monomers were added to the existing CDC pore to increase the pore size. Do the authors exclude the presence of pre-pores for MAC?

Introduction/Results/(Discussion)

From the introduction it is not very clear, whether a target membrane or the binding of the other subunits like C5-8 is needed for C9 oligomerization.

Supp. figure 1b suggests that in the soluble form of C9 the interface between the monomers is blocked by the tip of "TMH1" while the text (end of page 4 and beginning of page 5) suggests that TMH1 "would" anticipate additionally in the blockage by the HTH, which on the other hand is not even shown in the schematic drawing.

Suppl. figure 1b: It looks like, that the largest portion of the MAC beta-barrel is outside the membrane, and that the MAC interacts with the membrane only via the tips of the beta-strands.

Sequence alignment, especially of the regions involved in the self-inhibition of C9 in absence of the target membrane would be helpful.

Figure 1a lacks of clarity as the "unstructured" loops of the molecule make it difficult to follow the main chain, even with the different colors. The figure will become clearer using "cartoon smooth loops" in pymol (and manual secondary structure assignment as the secondary structure recognition of pymol often doesn't work optimally. It would also be helpful to show the "elongation surface" in the real structure.

The term "trans-membrane beta hairpin" for the region inserting the membrane is only correct for the pore form of MAC, but it is wrong to use this for the crystal structure or the non-membrane inserted form. Also the abbreviation "TMH" is misleading for the general readership as it can be mixed up with the more common term "trans-membrane helix".

The hemolytic assay with the mutant tells that the C9mut is inhibited in hemolytic activity. In combination with the wild type C9 it reduces the activity. But do the authors completely exclude the interaction of two or more C9 mutants? Suppl. figure 7 shows that the single mutants also slightly inhibit the hemolytic activity. Do they block the interface (and the polymerization) as well?

From the cryo-EM structure it is not really clear which region in the beta-barrel is defined as the trans-membrane region. Is the density for the amphipol defined in the map? In this context the surface charge distribution in suppl. figure 6 shows no clear hydrophobic patches towards the tips

of the beta-hairpins.

The arrangement of the HTH motif looks similar to the HTH found in the CDC pore structures. Is this motif conserved in the sequence of MACPF/CDC? Shows the Interface between two neighboring HTHs charge and surface complementarity?

The authors should show in one of the suppl. figures the density map for characteristic residues in the membrane region and in the region of the beta-sheet that is much better defined in the upper part. The map shown in figure 3d is not very informative.

Showing the dimensions of the pore complex in figure 3a would be helpful for the reader.

How is the lipid removed during the sequential insertion of the beta-hairpins?

Suppl. Figure 3: It seems, that some regions of the protein, including the membrane, are removed for clarity. If this is true, it should be stated in the figure legend.

Mat&Meth

Suppl. Table 3 shows the statistics for three datasets collected at a wavelength of $\sim 1.204\text{\AA}$ while the statistics for the dataset collected at 9674 eV showing the absence of the anomalous signal for Zn is missing.

Reference to "Table 1" in page 11 seems to be wrong.

Did the authors try to solve the structure by molecular replacement using one of the structures they used in Suppl. Fig. 3 (C6, C8 α/β).

Are the Asn sites as well mutated in the mammalian cell expressed C9 as in the murine C9 used for crystallization?

A table summarizing the cryo-EM data is missing.
Best.

Key:

Highlighted in yellow are the original reviewer comments

Our response is in italics.

Red text are the actual changes made to the manuscript.

Reviewer #1 (Remarks to the Author):

This manuscript by Spicer, Law et al. describes structural studies of monomeric C9 by crystallography and pore-forming polyC9 by cryo-EM. The structural work is done beautifully and the authors are congratulated with their important achievements.

However, I find it hard to follow their reasoning in terms of a mechanistic model in the sequential process of adding a monomer to the growing ring of polyC9. One difficulty lies in Fig. 1 b and c. The text mentions a “striking difference”. This is simply not clear to me from Fig. 1. Figure 3c does to some degree a better job of showing how the structure alters upon polymerisation.

Figure 1c has been replaced with an improved version to reflect the reviewers concern.

Next, an important aspect of the mechanistic model is the disulphide trapped variant of C9, which locks TMH1. Though, by itself the resulting data are interesting, the data could be interpreted in multiple ways. Clearly, a variant that is unable to form the two beta-hairpins presents in possibly multiple ways a corrupted ending making it harder to bind a next copy. To show that a structural element in monomeric C9 prevents polymerisation, would require mutants in which this element is removed or altered such that polymerisation proceeds more readily.

Despite extensive efforts, we (and others) have found that mutations and / or deletions cannot be readily made in the TMH1 region since this region appears to play a key role in the correct folding of C9 (see for example Weiland, Qian and Sodetz; Mol Immunol, 2015).

Accordingly, we designed disulphide bond trapped variant that could undergo a regain of function (upon addition of reducing agent). The ability of the molecule to regain function demonstrates the molecule is properly folded (Figure 2a). Further, the competition experiments (Fig 2c; together with the structural data) support strongly the idea that one molecule of the disulphide locked C9 can still bind to the C5b-8 complex and that addition of further C9 molecules requires reduction of the disulphide bond.

Nevertheless, the structural data themselves present hallmarks to the field. I would suggest to improve, in particular, Fig 1b/c and rephrase the manuscript w.r.t. its mechanistic interpretation.

We have rephrased the manuscript to make it clearer and improved Figure 1b/c.

Minor comments:

The abstract reports 2.3 angstrom resolution for the X-ray structures, whereas in the text this is 2.2 angstroms.

The abstract has been changed from stating 2.3 Å to 2.2 Å.

The gamma character is not correctly printed in the supplement file.

This has been corrected in the supplementary file.

Reviewer #2 (Remarks to the Author):

This manuscript describes the first high-resolution x-ray crystal structure of complement C9 and additionally a high-resolution cryo-EM structure of polyC9. Lack of C9 structural information has been a major obstacle to understanding details of how the MAC assembles and interacts with membranes. The authors work is highly significant in that it extends previous hypotheses about MAC assembly based on C6 and C8 structures and it provides new insight into how C9 can exist independently in monomer form yet self-polymerize when associated with the MAC. The authors' conclusions regarding the role of TMH1 in preventing C9 self-polymerization are plausible based on differences from structures of C6, C8a and C8b. Importantly, these conclusions are supported by experiments using mutants in which TMH1 is locked to a core β -strand. Determining the C9 structure alone is a major advance in the complement field but equally important are the mechanistic implications with respect to understanding polyC9 and MAC formation. These results will have a major impact on the area of complement.

The authors should identify the species of C9 as "mouse" in the title and/or abstract.

We have changed the abstract accordingly.

Reviewer #3 (Remarks to the Author):

The manuscript by Spicer et al. describes the monomeric X-ray structure of the immune effector Complement component 9 (C9) at 2.3Å and the cryo-EM structure of the polymeric C9 assembly (polyC9) of the membrane attack complex (MAC) at 3.9Å resolution. By comparison of the monomeric X-ray structure to the cryo-EM structure of the polymeric C9, the authors reveal molecular movements of important regions (HTH) in the molecule that appear to be required for assembly of the pore complex and subsequent pore formation. By mutating opposing residues in the central beta-sheet region and “TMH1” to cysteines and crosslinking them as disulfide bridges in the absence of DTT they verified that the movements they propose are required for functionality of C9 to form the pore complex.

The authors performed their experiments properly and present their results very briefly. Nevertheless, the conclusions were supported by the data. The manuscript fulfills all the requirements and therefore suitable for publication in Nature communications after the following points are addressed, clarified or discussed.

As the text structure suggests, the manuscript seems to be written very condensed in the introduction and results (for Nature?). The discussion is somehow squeezed into the text, which to my knowledge is not necessary for Nature communications. Therefore the manuscript would benefit from an expansion and broader description and discussion of the results.

We have tried to keep the manuscript as concise as possible. However, we have expanded the introduction (see below) and at the appropriate points we have now commented about some of the interesting points this reviewer made. In particular a discussion in with respect to pre-pores formation.

Abstract:

Last sentence of the abstract: How the authors distinguish the MAC pore formation by sequential addition of C9 monomers from that of the CDC pore formation by the “growing-pore mechanism” where the CDC monomers were added to the existing CDC pore to increase the pore size. Do the authors exclude the presence of pre-pores for MAC?

This is an interesting question. AFM and biochemical data support strongly the idea of transient pre-pore formation for the CDCs. In contrast, for the MAC it is suggested that sequential insertion takes place. Our data is consistent with TMH1 being released from the elongation face, an event that is most consistent with this region inserting into the membrane. However, we cannot completely exclude formation of an intermediate pre-pore-like state, where TMH1 has moved to permit C9 binding, but has not fully inserted into the membrane. We do not, however, have any evidence for formation of a partially released pre-pore such as is observed for pleurotolysin. We now comment on this interesting point in the discussion.

Introduction/Results/(Discussion)

From the introduction it is not very clear, whether a target membrane or the binding of the other subunits like C5-8 is needed for C9 oligomerization.

We have changed this section to:

In contrast, the MAC is unusual in that it is initiated by a non-MACPF domain protein, C5b, which then allows the sequential binding of single units of the MACPF-domain containing proteins C6, C7 and C8 complex (C8 α :C8 β :C8 γ). This assembly then allows the binding of multiple units of C9 into a ring shape and final pore formation (Supp Fig 1a). An interesting

feature of C9 is that it can also form a homogenous ring, called polyC9, that has only ever been observed in vitro.

Supp. figure 1b suggests that in the soluble form of C9 the interface between the monomers is blocked by the tip of "TMH1" while the text (end of page 4 and beginning of page 5) suggests that TMH1 "would" anticipate additionally in the blockage by the HTH, which on the other hand is not even shown in the schematic drawing

We have changed Figure 1b/c to reflect this point.

Suppl. figure 1b: It looks like, that the largest portion of the MAC beta-barrel is outside the membrane, and that the MAC interacts with the membrane only via the tips of the beta-strands.

The amphipathic membrane spanning region is, indeed, located at the base of the barrel. The rest of the barrel is proposed to span lipo-polysaccharide moieties.

The green sticks in Supp Fig 1b representing the membrane inserting regions have been made to look like they are inserted across the membrane rather than the tips touching the membrane.

Sequence alignment, especially of the regions involved in the self-inhibition of C9 in absence of the target membrane would be helpful.

We have added the sequence alignment of the TMH1 region to Supp Figure 2 and changed the figure legend accordingly.

Figure 1a lacks of clarity as the "unstructured" loops of the molecule make it difficult to follow the main chain, even with the different colors. The figure will become clearer using "cartoon smooth loops" in pymol (and manual secondary structure assignment as the secondary structure recognition of pymol often doesn't work optimally. It would also be helpful to show the "elongation surface" in the real structure.

Figure 1a has been adjusted accordingly.

The term "trans-membrane beta hairpin" for the region inserting the membrane is only correct for the pore form of MAC, but it is wrong to use this for the crystal structure or the non-membrane inserted form. Also the abbreviation "TMH" is misleading for the general readership as it can be mixed up with the more common term "trans-membrane helix".

Historically, throughout the field the term TMH-1 and -2 has been used extensively to describe the membrane spanning regions, whether they be in a folded, or membrane inserted state. While we completely understand the reviewers point, given the universal use of these terms throughout the CDC/MACPF field we would like to retain this nomenclature in the work.

The hemolytic assay with the mutant tells that the C9mut is inhibited in hemolytic activity. In combination with the wild type C9 it reduces the activity. But do the authors completely exclude the interaction of two or more C9 mutants?

Our structural data strongly suggests that the most likely that only a single C9mut molecule can bind to the C5b8 complex due to the massive steric hindrance of the TMH1 region.

Suppl. figure 7 shows that the single mutants also slightly inhibit the hemolytic activity. Do they block the interface (and the polymerization) as well?

We consider a change of LD50 of one order of magnitude to be significant – we thus argue that the small differences in activity exhibited by the single variants are within the error of the experimental approach (Fig 2c). For this reason we designed mutants such that we can observe a complete loss or regain of activity (Figure 2b) observe a clear trend of loss of activity in a dose dependent manner (Fig 2c).

From the cryo-EM structure it is not really clear which region in the beta-barrel is defined as the trans-membrane region. Is the density for the amphipol defined in the map? In this context the surface charge distribution in suppl. figure 6 shows no clear hydrophobic patches towards the tips of the beta-hairpins.

The model presented in the manuscript model was built into the maps using de novo model building and real space refinement (Methods section). Accordingly, only the regions that were well defined in electron density were built. The transmembrane region and the amphipols were not built as these regions had lower local resolution (Supp Fig 5).

We have added to the Figure 3 legend the following statement:

The model excludes the membrane spanning region due to the lower resolution of this portion of the map (Supp Fig 5a).

The arrangement of the HTH motif looks similar to the HTH found in the CDC pore structures. Is this motif conserved in the sequence of MACPF/CDC? Shows the Interface between two neighboring HTHs charge and surface complementarity?

The HTH structure is broadly conserved, however, little sequence identity is observed.

The authors should show in one of the suppl. figures the density map for characteristic residues in the membrane region and in the region of the beta-sheet that is much better defined in the upper part. The map shown in figure 3d is not very informative.

See Sup fig 5 - the membrane penetrating region is at relatively low resolution, hence we have not modelled this region. We have retained showing density in Figure 3d to illustrate the quality of the map in this area.

Showing the dimensions of the pore complex in figure 3a would be helpful for the reader.

The inner and outer diameter of the pore have been added to figure 3a. More detailed dimensions are shown in Supp Fig 5d. We have added the following text to the Supp Fig 5 figure legend:

d) Unsharpened map with the final atomic model excluding the TM region which were not modelled due to lower resolution. The dimensions are also shown (220 Å denotes the outer-most dimension and 120 Å denotes the inner most dimension of the pore).

How is the lipid removed during the sequential insertion of the beta-hairpins?

This is not known. However, Leung et al., eLife, 2014, speculate that the lipid “blebs” off or is shed in response to pore formation.

Suppl. Figure 3: It seems, that some regions of the protein, including the membrane, are removed for clarity. If this is true, it should be stated in the figure legend.

This has been rectified by adding the following to the Supp Fig 3 legend:

For clarity, the TMH1 and TMH2 regions of the MACPF domain of the models have been removed. The surrounding domains have also been excluded.

Mat&Meth

Suppl. Table 3 shows the statistics for three datasets collected at a wavelength of ~1.204Å while the statistics for the dataset collected at 9674 eV showing the absence of the anomalous signal for Zn is missing.

This is addressed in Supp Table 3b.

Reference to “Table 1” in page 11 seems to be wrong.

This has been corrected from Table 1 to Supp Table 3a.

Did the authors try to solve the structure by molecular replacement using one of the structures they used in Suppl. Fig. 3 (C6, C8α/β).

Yes, molecular replacement was attempted using the MACPF domain of C6 (PDB ID 3T5O) and with both MACPF domains of C8 (PDB ID: 3OJY). Neither MR experiments were successful. To clarify this point we have made the following change in the methods section:

Molecular replacement was attempted using the MACPF domain of C6 (PDB ID 3T5O) and with both MACPF domains of C8 (PDB ID: 3OJY). None of these experiments were successful.

Are the Asn sites as well mutated in the mammalian cell expressed C9 as in the murine C9 used for crystallization?

No, in the electron microscopy studies of the human form of polyC9 the wild-type version with the two native N-glycans was used. Unpublished studies of human C9 without N-glycans shows that the polyC9 structure formed is essentially the same. In the murine form, the Asn sites were mutated specifically for crystallisation as the murine C9 with N-glycans failed to crystallise. To clarify this point we have made the following change in the methods section:

Mammalian cell expressed human C9 (including the two native N-glycans) was buffer exchanged...

A table summarizing the cryo-EM data is missing.

A table summarizing the cryo-EM data has been added to the supplementary section as Supp Table 4.

REVIEWERS' COMMENTS:

Reviewer #3 (Remarks to the Author):

This version of the manuscript is improved according to the comments of all reviewers and the authors responded to all comments I made. I have no further issues that need to be addressed for the acceptance of the manuscript.

One last comment to the authors response:

"Historically, throughout the field the term TMH-1 and -2 has been used extensively to describe the membrane spanning regions, whether they be in a folded, or membrane inserted state. While we completely understand the reviewers point, given the universal use of these terms throughout the CDC/MACPF field we would like to retain this nomenclature in the work."

Scientist should be able to correct old-established (wrong) terms.

Best

Key:

Highlighted in yellow is the reviewer communication

Our response is in italics.

Reviewer #3 (Remarks to the Author):

This version of the manuscript is improved according to the comments of all reviewers and the authors responded to all comments I made. I have no further issues that need to be addressed for the acceptance of the manuscript.

One last comment to the authors response:

"Historically, throughout the field the term TMH-1 and -2 has been used extensively to describe the membrane spanning regions, whether they be in a folded, or membrane inserted state. While we completely understand the reviewers point, given the universal use of these terms throughout the CDC/MACPF field we would like to retain this nomenclature in the work."

Scientist should be able to correct old-established (wrong) terms.

We understand the reviewer's point (and we have defined TMH to improve clarity), but feel that this paper is not the correct place to address this nomenclature issue.